# Assessment of a Fully Renewable System for the Total Decarbonization of the Economy with Full Demand Coverage on Islands Connected to a Central Grid: The Balearic Case in 2040

Yago Rivera [1], David Blanco [1], Paula Bastida-Molina [1,2,*] and César Berna-Escriche [1,3]

1 Instituto Universitario de Investigación en Ingeniería Energética (IIE),
Universitat Politècnica de València (UPV), Camino de Vera 14, 46022 Valencia, Spain; yaridu@upv.es (Y.R.); dablade@upv.es (D.B.); ceberes@iie.upv.es (C.B.-E.)

2 Departamento de Ingeniería Eléctrica, Universitat Politècnica de València (UPV), Camino de Vera 14, 46022 Valencia, Spain

3 Departamento de Estadística, Investigación Operativa Aplicadas y Calidad,
Universitat Politècnica de València (UPV), Camino de Vera 14, 46022 Valencia, Spain

* Correspondence: paubasmo@upv.es; Tel.: +34-963879245

**Abstract:** The transition to clean electricity generation is a crucial focus for achieving the current objectives of economy decarbonization. The Balearic Archipelago faces significant environmental, economic, and social challenges in shifting from a predominantly fossil fuel-based economy to one based on renewable sources. This study proposes implementing a renewable energy mix and decarbonizing the economy of the Balearic Islands by 2040. The proposed system involves an entirely renewable generation system with interconnections between the four Balearic islands and the Spanish mainland grid via a 650 MW submarine cable. This flexible electrical exchange can cover approximately 35% of the peak demand of 1900 MW. The scenario comprises a 6 GWp solar photovoltaic system, a wind system of under 1.2 GWp, and a 600 MW biomass system as generation sub-systems. A vanadium redox flow battery sub-system with a storage capacity of approximately 21 GWh and 2.5 GWp power is available to ensure system manageability. This system's levelized electricity cost (LCOE) is around 13.75 cEUR/kWh. The design also incorporates hydrogen as an alternative for difficult-to-electrify uses, achieving effective decarbonization of all final energy uses. A production of slightly over $5 \times 10^4$ t$H_2$ per year is required, with 1.7 GW of electrolyzer power using excess electricity and water resources. The system enables a significant level of economy decarbonization, although it requires substantial investments in both generation sources and storage.

**Keywords:** renewable energy; wind generation; solar photovoltaic generation; mega-batteries; sub-grid interconnections; mainland grid connection; hydrogen production; exploratory analysis; sensitivity analysis

## 1. Introduction

For the last decades, climate change has become one of the most disturbing global problems. The high amount of greenhouse gases (GHG), mainly $CO_2$ emissions, sent to the atmosphere is leading to environmental destruction, whose consequences could be very negative for nature and, therefore, for our society [1,2].

Of these emissions, 77% correspond to the Energy sector [3], which in turn includes electricity production. This sector is essential nowadays since electricity supply provides the key element for the economic, social, and environmental development [4] of different countries. Thus, the carbon intensity (CI) of the technologies involved in electricity generation directly matches the $CO_2$ emissions of this sector and its environmental degree [5]. This important issue (technologies involved in the electricity mix) is what most of the

current legislation point to. The current European Union (EU) target establishes a 32% renewable sources participation in the electricity mix by 2030 [6]. However, REPowerEU, presented in May 2022, highlights the importance of accelerating the transition towards clean energy and gradually reducing imports from Russia. It proposed increasing the renewable sources share by up to 45% by 2030. These measures correspond to the "Fit for 55" package, which aims to reduce EU GHG emissions by at least 55% by 2030 [7]. Moreover, the Paris Agreement established a basis for limiting global warming to 1.5 °C, which requires a profound decrease in GHG from its peak in 2020 down to zero by the end of this century [8].

Most of the European countries are working to fulfill these requirements. Some of these countries include islands territories, which due to their inherent features, turn out to be challenging and beneficial, at the same time, for the energy transition. On the one hand, they are isolated territories, dependent nowadays mostly on imported fossil fuels for electricity generation as well as on their own resources, if any. On the other hand, electricity interconnection with their mainland sometimes is not possible due to the long distances that take place [9]. This situation aims to enhance the opportunity of using the own islands' renewable resources, mainly solar photovoltaic (PV) or wind, to promote hybrid renewable penetration for their electricity supply [10]. Interconnections among the closer islands have traditionally been used to compensate for renewable energy sources (RES) variability. For instance, Alves et al. analyzed the interconnection between Pico and Faial in the Azores (Portugal), concluding that RES penetration can increase by 50% by 2030 [11]. Moreover, stand-alone grids for islands with high RES penetration have been studied in this literature review. Vargas-Salgado et al. focused on Gran Canaria ( part of the Spanish Canary Islands), which is not interconnected with the Spanish peninsula, determining that solar PV and wind technologies, supported by lithium-ion batteries and pumped storage systems, were capable of covering the island's demand by 2040 [12]. Although pumped storage systems arise as very flexible back-up resources, their implementation highly depends on the geographic conditions of the land since high height differences are needed for hydropower energy. Therefore, researchers analyzed other back-up technologies, and very few explored novel cutting-edge energy vectors such as hydrogen. Berna-Escriche et al. demonstrated that hydrogen can be considered a real alternative for future scenarios since new electric items are gradually being included in society: heavy road transport, maritime, and air transport. They focused on Gran Canaria, considering three possible scenarios where electricity surplus was in the range of 2.3–4.9 TWh/year, concluding that it would be possible to produce $3.5 \times 10^4$ to $7.68 \times 10^4$ t of $H_2$/year [13].

Furthermore, other researchers have also analyzed how the connection with the mainland (when possible) increases the security of supply for islands' electricity supply. Zafeiratou et al. analyzed the Greek islands' interconnection with their mainland, concluding that loss of load probability and unserved energy are eliminated, whilst GHG emissions are reduced by 73% by 2040 since RES share could be increased [14].

From this review, some research gaps have been identified. Firstly, previous authors only considered two approaches: islands with completely off-grid power systems or with connections to the mainland. Moreover, considering the forecasted increase in electricity demand in the islands, the use of hydrogen remains mainly unexplored. Thus, this research paper covers both research gaps: it presents a power system for islands with high RES penetration, supported by batteries and hydrogen, together with an interconnection with the mainland. Specifically, the Balearic Islands (Spain) have been chosen as a case study.

The Balearic Islands are the Spanish autonomous community with the highest foreign energy dependence and the lowest implementation of renewables. Coal and oil are a particularly important part of the fossil fuels used in electricity generation. Additionally, the ratio of private cars per inhabitant is higher than the Spanish average. In addition, this archipelago has the highest tourist intensity index of all of the world's island territories. All this makes the Balearic Islands the Spanish territories with the highest emissions.

However, on the other hand, its insularity is also an opportunity for the energy transition toward a sustainable model. This study presents the main contributions of a proposed power generation, storage, and hydrogen production system for the Balearic Islands. This system demonstrates the feasibility of achieving a renewable energy supply using zero GHG emission sources, such as wind, solar photovoltaic, biomass, and exchanges with the mainland grid. It incorporates flexible technologies to optimize the utilization of solar and wind generation and addresses seasonal variations and challenges. The economic analysis highlights the substantial investment required for system implementation. Overall, this study provides valuable insights into the potential for achieving a sustainable and low-carbon energy future in the Balearic Islands, characterized by the use of renewable energies, energy efficiency, efficient utilization of natural resources through the introduction of a circular economy, and sustainable development and mobility.

To implement the above scenario, the software developed by the National Renewable Energy Laboratory (NREL) (Golden, CO, USA), called Hybrid Optimization Model for Multiple Energy Resources (HOMER) [15], is used. According to [16], HOMER stands out from other hybrid renewable energy modeling tools since it is the software that can analyze the most inputs and outputs and that can cover the most energy technologies. The main criterion followed by the software is economical since the code estimates the optimal size of a system based on the investment to be made and the amortization based on the energy sources to be installed, i.e., the program presents a series of results of possible scenarios that meet the required conditions ordered by their LCOE (Levelized Cost of Energy). It should be noted that the HOMER software has been widely used in various simulations of hybrid renewable systems, which have different generation sources and storage technologies [17,18].

To achieve the objectives described above, this paper is organized as follows: Section 2 describes the electricity system of the Balearic Archipelago; Section 3 focuses on the description of the methodology developed to carry out the proposed analysis; Section 4 briefly describes the characteristics and the demanded information necessary for all the systems needed to perform the simulations in the desired horizon, both for generation and storage technologies, as well as describing the demands of the two types of energy vectors used, electricity and hydrogen. The main results of the simulations performed are presented in Section 5, with the corresponding analysis and discussion. Some possible future work is also mentioned in this section. Section 6 is devoted to summarizing the conclusions of the present study with regard to the generation system in the scenario considered.

## 2. The Balearic Energy System

A detailed description of the final energy consumption forecasts for a fully electrified system by 2040 for the Balearic Islands is described in the Monitor Deloitte report [19] and shown in Figure 1. This research suggests an electricity demand of approximately 10.3 TWh per year, which would mean assuming an annual increase of approximately 3%. This expansion would be mainly due to the electrification of transport and economic growth. Another 1.7 TWh per year would have to be added to this amount if those end uses of energy that are difficult to electrify are covered, mainly heavy land and maritime transport, as well as those industrial uses with high energy requirements. For these uses, hydrogen currently seems to be the most advantageous alternative, which means employing approximately $50 \times 10^3$ tons of $H_2$ per year to cover the forecasted 1.7 TWh of non-electrifiable energy end-uses. According to the Monitor Deloitte's report [19], to achieve this completely decarbonized electricity system by 2040, considering a net interconnection capacity with the Peninsula of at least 400 MW, about 5 GW of installed renewable capacity and approximately 13–14 GWh of storage capacity would be necessary.

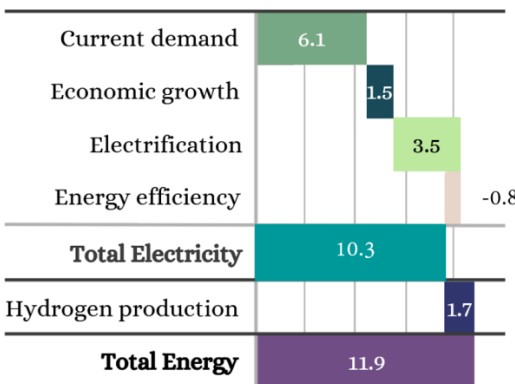

**Figure 1.** Forecast of the demand (TWh/year) in the Balearic Islands for 2040 (adapted from [19]).

Analyzing in detail the different characteristics of the energy generation and consumption system of the Balearic Islands, it can be said that this system presents a series of general behaviors of this type of system, but it also presents some particularities. In this regard, an examination of the system reveals a series of key observations [19–22].

Decarbonization of the economy is the solution to achieve the ambitious goal of eliminating GHG emissions. To this end, maximizing the use of electricity as the primary energy vector in most energy end-uses is crucial. Additionally, to accomplish the decarbonization of the last approximately 10% of energy end-uses, which relies on electricity generated from RES, a comparable investment to electrifying the previous 90% is required [19]. Therefore, it is essential to implement a technology that offers affordable backup solutions. Hydrogen emerges as a promising energy vector to fulfill this role.

Regarding final energy consumption (2019 energy balance data), it is estimated that more than 75% comes from fossil fuels or oil derivatives and another 25% from electricity generation. Within the electricity system, approximately 5% is currently sourced from renewable energy produced within the Balearic Islands. The transport sector is the largest consumer, accounting for over 60% of final energy consumption and being responsible for more than 50% of total GHG emissions in non-peninsular territories (around 3.5 $MtCO_{2eq}$ per year in the Balearic Islands). Private passenger road transport represents 80–85% of the sector's energy consumption, while public transport (6%), road freight (7–9%), inter-island maritime transport (1–4%), and inter-island air transport (1–2%) have significantly lower consumption rates [22].

The residential and services sectors, on the other hand, contribute to around 30% of energy consumption, displaying a high degree of electrification ranging from 70% to 80%. Industrial, agricultural, and other sectors represent less than 10% of the overall energy consumption [22]. Shifting from private to public transport has the potential to reduce GHG emissions by approximately 70% compared to conventional buses, but it requires improvements in service supply and quality, as well as restrictions on private vehicle use. The adoption of electric or hydrogen buses as alternatives to traditional public transport modes is still uncertain [19].

The Electric Vehicle (EV) offers a more economical solution than conventional vehicles, with a cost reduction of 10% in terms of full cost and sufficient autonomy for island environments. On average, EVs emit 55% less greenhouse gases compared to conventional vehicles [5,19], contributing to sustainable economies. However, barriers such as inadequate public charging infrastructure and higher upfront acquisition costs hinder widespread EV adoption. In the rental sector, which plays a significant role in the islands, EVs are not necessarily more cost-effective due to the higher initial vehicle costs borne by rental companies, as fuel expenses are typically paid by customers. Thus, rental businesses should modify their pricing models to effectively leverage EV rentals in these territories.

The residential sector, responsible for approximately 1–3% of direct GHG emissions, focuses on heating and domestic hot water (DHW) consumption [19]. Decarbonizing this

sector necessitates deploying efficient heat pump technology for heating and DHW, given its capacity to significantly reduce emissions. Additionally, promoting self-consumption holds advantages for island territories. In the service sector, accounting for approximately 2–4% of direct GHG emissions, emissions primarily stem from restaurants, tourist accommodations, commerce, and offices. Notably, air conditioning (including heating systems) and DHW constitute a substantial portion (50–70%) of energy consumption in hotels and accommodations [19]. Addressing this challenge entails replacing inefficient electrical equipment and thermal systems fueled by natural gas or petroleum products with heat pumps. Among industrial sectors, some demonstrate high potential for electrification using current technologies (textile, leather and footwear, manufacturing, food, beverages, tobacco, chemicals, etc.) with an average electrification rate of 34%. However, certain industries rely on high-temperature processes (clinkerization in cement, high-temperature furnace processing in glass, calcination and heat treatment in metal products). Decarbonizing these high-temperature industries may require exploring technologies such as hydrogen [19].

The Balearic system is already interconnected with the mainland through the Romulo Project, which currently provides a 400 MW high-voltage submarine connection at $\pm 250$ kV. This interconnection comprises two power cables and a third return cable, extending approximately 237 km with a maximum depth of 1485 m. However, future infrastructural plans anticipate expanding this interconnection by the end of the decade, allowing for an increase of up to 650 MW. This high-power interconnection with the mainland is crucial as it ensures a stable power supply, addressing the inherent unpredictability and variability of RES [23]. Moreover, the interconnection capacity between the islands is equally significant. In the past, the Balearic system consisted of two small and electrically isolated sub-systems: Mallorca-Menorca and Ibiza-Formentera. The islands of Mallorca and Menorca were connected by a 132 kV submarine circuit, while Ibiza and Formentera had two 30 kV submarine links. As part of the Romulo project's second phase, an electrical connection was established between Mallorca and Ibiza, featuring a 126 km long three-pole double link operating on alternating current at 132 kV voltage and $2 \times 100$ MW of power. This remarkable AC link holds the distinction of being the world's longest and deepest of its kind, traversing seabed depths of up to 800 m [20].

The average cost of generation in the Balearic Islands is significantly higher than on the mainland, reaching around 150 €/MWh compared to 55 €/MWh, primarily due to the higher fuel costs associated with existing generation technologies and the challenges of achieving economies of scale. Despite the possibility of a robust interconnection capacity, the Law on Climate Change and Energy Transition of the Balearic Islands sets a minimum target for island-based energy production, aiming to reach 70% of total energy consumption by 2050 [22]. This emphasizes the importance of local energy generation and self-sufficiency.

Solar PV generation is a better fit with storage, as solar energy is more predictable compared to wind energy, which can experience low wind periods requiring larger storage systems. For example, estimates for the Balearic system suggest that a generation mix of 50% solar PV and 50% wind would require approximately ten times more storage capacity than a mix of 90% solar PV and 10% wind [18]. On the other hand, biomass plays a crucial role as a backup generation system due to its reliability and fast activation, making it the only suitable renewable source for backup purposes. Together with storage systems and interconnection with the mainland, biomass ensures the necessary supply reliability that wind and solar photovoltaic sources cannot offer due to their inherent variability.

To ensure the security of supply during interconnection failures and emergencies, an emergency backup capacity should be installed on the islands. This can be achieved through biomass gasification plants or by considering the installation of technologies for the potential re-electrification of hydrogen produced for other applications. Maximizing the biomass exploitation capacities of the islands and installing significant generation capacities are essential to serving as an emergency backup system, which can also be utilized during regular operation when technically and economically viable. Additionally, alternative solutions such as fuel cells, turbines, or hydrogen engines could be explored.

Land use may be another key factor in the Balearic Islands. Although, in the current case, it should not be a relevant barrier if the full potential for self-consumption is exploited, the estimations are above 2.3 GW considering conservative roof occupations. Whereas if rural land is considered for the installation of intensive exploitation plants, the installable power far exceeds the power needs (forecasts of more than 70 GW) [20].

The interconnection between islands and the mainland is effectively complemented by the renewable generation and storage system. Renewable generation (primarily solar PV) operates at maximum capacity during the central hours of the day, so storage captures excess generation for use at night. Interconnection supports storage in its task of meeting demand during nighttime hours. This support means that the more interconnection capacity there is, the less storage is needed to meet demand. Additionally, biomass generation is available as a back-up system, in case of need, due to failure of the interconnection system or because it is not sufficient.

The development of demand management, the ability to shift demand towards generation, is an additional solution that can significantly reduce the required storage capacity and should be explored. This entails the introduction of an electricity tariff and hourly price signals that incentivize consumption during peak renewable production hours, specifically the central hours of the day in the case of a predominantly solar mix. Additionally, mechanisms must be developed to enable the System Operator to manage demand when necessary, in parallel and coordination with how the System Operator would manage storage. These mechanisms will vary for each consumer type and may include demand aggregators, systems for managing connected electric vehicles, or the evolution of interruptibility for large consumers. An appropriate regulatory framework and operational procedure would be necessary to ensure clear and transparent management of this service by the System Operator.

## 3. Methodology

The HOMER code estimations have been carried out following an established methodology, which consists of two main parts [15]. The first one is the introduction of the input data required for the code to perform the simulation and, after it, the implementation of the different stages presented in Figure 2. The input data include various types of information, such as the annual energy demand or its projections for future scenarios, for both electricity and hydrogen; the technical specifications and costs of the generation system under consideration (in this case, wind, solar PV and biomass power facilities); the technical features and costs of the storage system (mega-batteries and/or reversible pumping, if both are necessary, only one, or if either is not available); the availability of the energy resources for each generation system (wind, solar PV and biomass resources at the selected locations); and other additional economic data (such as the annual interest rate and the project lifetime). Using HOMER software, these input data are processed to find the most efficient and cost-effective combination of generation and backup systems that can meet the energy demand of the system. The software also determines the optimal values for parameters such as the nominal power, the power output from each system, the storage capacity, etc. Moreover, HOMER provides economic indicators such as LCOE, initial capital cost, net present cost (NPC), payback, and Internal Rate of Return (IRR). The methodology selects the best option based on these economic indicators while ensuring that $CO_2$ emissions are zero.

The main objective of the code is to meet economic criteria, that is, to find the optimal balance between the size of the generation and storage facilities and the demand needed to minimize the system costs. That is, the optimal solution is the one that has the lowest cost for the system during its whole lifetime. To estimate the cost of the system, the current economic data has been used, and it has been assumed that the cost variation of the technologies will remain constant until 2040. This assumption is reasonable because the majority of the technologies that have been used are mature and well-established.

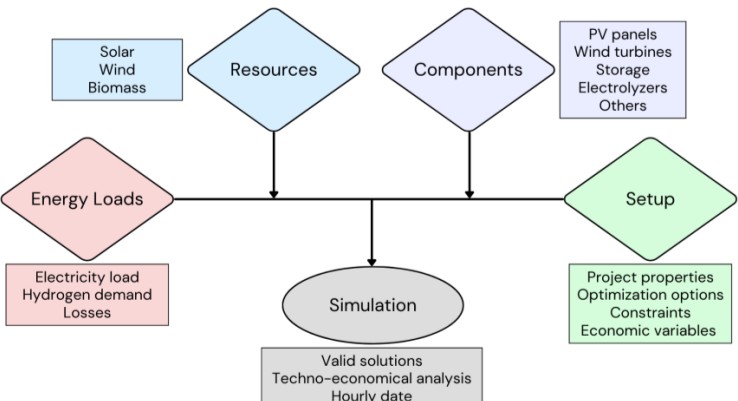

**Figure 2.** Schematic overview of the HOMER software information of inputs and outputs.

To simulate the operation of each system, the HOMER code performs an energy balance at every defined time step (usually hourly). Then, the software compares the energy demand at each time step with the energy that can be supplied by the generation system and determines how to operate the generators and when to use the storage system. After simulating all possible system configurations, HOMER provides a list of feasible systems ranked by their NPC. The optimal solution is the one with the lowest NPC, but other solutions can also be considered based on other criteria, such as lower total installed power, minimization of energy wastages, or a weighted combination of several factors. However, in this case, we follow the common practice of other researchers and select the global optimal solution as the best one [12,13].

The previously mentioned calculations have been carried out for the aggregate system. One optimal location for each one of the different generation and storage systems on the Balearic Islands has been used. For instance, locations near the highest consumption island (Mallorca) have been selected to install the wind turbines, so it does not need to oversize the interconnection between the different islands.

As detailed previously, the software uses different input data (wind and solar resources, annual energy demand, both electricity and hydrogen; technical specifications and costs of the electricity generation and storage subsystems, technical specifications and costs of the hydrogen production sub-system, as well as other additional economic data, among other required information) to make its estimates. Basically, the selection criterion is economic, i.e., it looks for the generation mix that covers the demand at a lower cost. However, it is possible to find the optimal solution based on other conditions. For example, it is possible to look for the mix that minimizes the excess energy of the system, the one with the lowest total installed power, or a compromise solution between several of them.

In addition, it is also worth mentioning that the code can also define the restrictions that are considered necessary; some of the restrictions considered are detailed below. For example, in this study, it has been considered that only the generation through self-consumption is convenient due to it being a tourist island; moreover, it is not convenient to install solar farms that produce landscape pollution, so a maximum value of this has been estimated. Along the same line, in wind power generation, only off-shore generation has been considered for the same reason, not to produce landscape pollution of the environment. A biomass gasification power generation has been implemented, considering the existing own resources in the islands, so the production of this generation system has been limited. On the one hand, storage technology, by means of reversible pumping, has not been considered either because there are no suitable sites or previously built infrastructures that could be used; although it is true that the construction of these systems from scratch could be considered it would obviously be more expensive, furthermore, this is also due to the restriction of landscape pollution mentioned above.

The code processes the different information provided to it so that it proposes a long list of options to meet the objective of covering the demands (electricity and hydrogen). Finally,

from this list, the system that presents the most efficient and cost-effective combination of generation/production and storage systems to meet the energy demand of the system with zero shortage is selected. Once the optimal mix has been selected, the different information related to the performance of the system is extracted, for example, the values of the nominal powers of each subsystem, the energies generated by each of them, together with their hourly evolution throughout the year, the evolution of the storage capacity, the existing energy surpluses, etc. In addition, HOMER provides economic indicators such as LCOE, initial capital cost, NPC, depreciation and inIRR, so that the system performance can also be estimated in economic terms.

### 3.1. The Balearic Demand and Electric System

In the proposed scenario, the entire archipelago's hourly demand has been computed by aggregating the hourly demand curves of all islands for the whole electricity uses (residential, commercial, industrial, public administration, hotels and other uses). For each island, the demand data from 2019 was taken as the starting point to project the demand for 2040. To achieve this, several factors described previously in Figure 1, including economic growth, electrification unrelated to electric vehicles (EVs), and energy efficiency, were incorporated. The 2019 values were scaled accordingly, and the result is depicted as the blue line in Figure 3, representing the projected demand for 2040 without the contribution of EVs. Subsequently, the contributions of EVs were introduced as separate hourly data with their own demand curve, represented by the green line in the same figure. The estimation of the EV contribution was derived from weighted averages of six EV recharging profiles for each island, including various charging locations such as homes, public roads, workplaces, hotel parking areas, shopping centers, and regular recharging points [24].

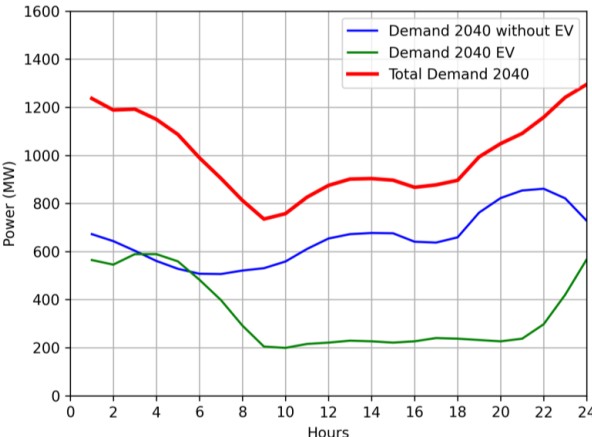

**Figure 3.** Electricity demand profile forecast of the Balearic Islands for 2040.

Notably, the EV contribution accounts for approximately 85% of the electrification portion [19]. The final estimated hourly average curve for the electric demand in 2040 is depicted as the red line in Figure 3. It displays a substantial increase, mainly attributable to the significant impact of the introduction of EVs.

The Balearic electricity system currently consists of a single electricity system connected to the mainland since 2012 by means of a 400 MW bipolar link with a return cable (Figure 4). The cable of this project (Romulo project) has an approximate length of 237 km, with a maximum depth of 1485 m (Figure 5). While in 2015, the second phase of the Romulo project was carried out, which interconnected the two sub-systems that existed until then, Mallorca-Menorca and Ibiza-Formentera. However, in order to consider the requirements foreseen for 2040, an estimate of the interconnection costs, powers, lengths and their associated losses have been taken into account since the significant increment in the forecasted demands for 2040 requires enhancement of the current facilities. The calculations consider the distances of the submarine cable, those of the undergrounding on

land from the substations to the coast, as well as the length of the submarine route, also considering the particular submarine orography of each case. As an example, Figure 5 shows the profile of the interconnection between the Iberian Peninsula and the island of Mallorca. Table 1 shows a summary of the fundamental characteristics of the connections considered. It should be noted that all of them are currently operational, although the extension of some of them has been considered to be necessary in order to be able to manage the inevitable increase in demand in the case of the electrification of the economy. Regarding the internal distribution network of the islands, it has been considered that the improvements to be made to the current ones would present a value of 359 million euro [25]. This value has been estimated from that of 2011-2021, a period of time in which the demand has increased in percentage terms similar to that estimated in the current study.

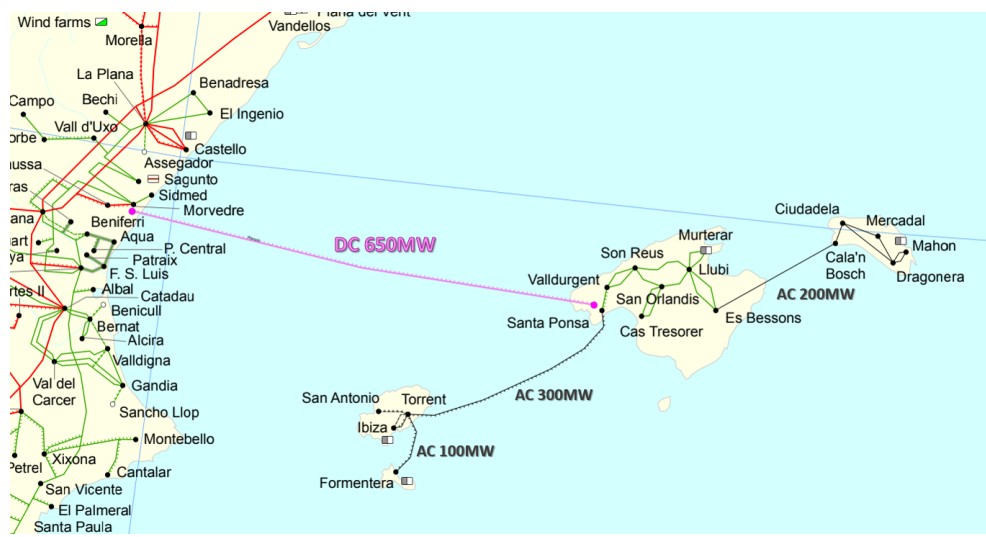

**Figure 4.** Proposed grid interconnection between the Balearic Islands and the Iberian Peninsula.

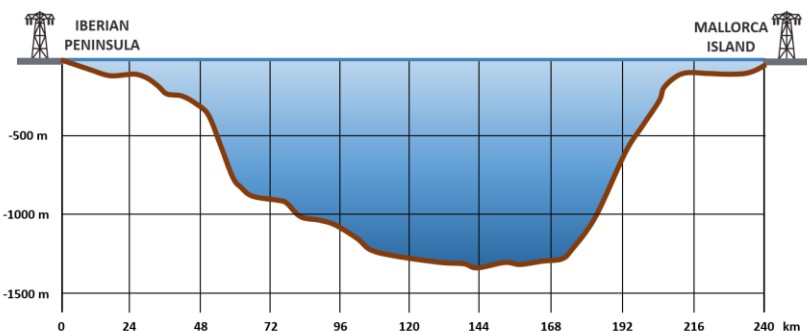

**Figure 5.** Profile of the interconnection between the Iberian Peninsula and Mallorca Island.

**Table 1.** Major characteristics of the Balearic interconnection system.

| Interconnection | Power (MW) | Length (km) | Cost (M€) [1] | Total Losses (%) [2] |
|---|---|---|---|---|
| Iberian Peninsula-Mallorca | 650 | 237 | 720 | 1.95 |
| Mallorca-Menorca | 200 | 55 | 293 | 1.21 |
| Mallorca-Ibiza | 300 | 126 | 417 | 1.50 |
| Ibiza-Formentera | 100 | 37 | 183 | 1.15 |

[1] Costs of all the interconnections have been estimated using the Qiblawei et al. [26] correlation, $C_{SM} = 7.172 \cdot P_{MW}^{0.5989} \cdot L_{km}^{0.1336}$. [2] Losses are estimated according to the conservative expression $(1 + 0.4 \cdot \text{Length}/100)$ [26].

In relation to the sizing of the interconnections, this has been done in such a way that they can allow energy exchanges between islands under any circumstances. To this end, the interconnections with the islands located at the ends of the grid (Menorca and

Formentera) allow the passage of the peak demand foreseen for them, plus a 10% safety margin, so that no extensions should be made over a long period of time (the demand in 2040 for the islands should be stabilized, once the changes produced by the decarbonization of the economy are implemented, mainly due to the increase in demand caused by EVs). For the island of Ibiza, the sizing of the interconnection allows the passage of its peak demand added to that of Formentera plus the 10% safety margin mentioned above. In Mallorca, the central island with a demand of approximately 75% of the total of the islands, most of the generation and storage systems will be concentrated, both electric and hydrogen, which will be described throughout the research. Thus, each island will have approximately a proportional share of each of the generation and storage systems. The situation in which energy exchanges between them should be carried out not continuously but mainly when it becomes necessary will occur as a result of an anomalous situation due to a breakdown or the occasional failure of a resource or storage capacity. A summary of the major characteristics of all interconnections is displayed in Table 1.

As it has been done for electricity, the shape of the hydrogen demand curves must be estimated by first analyzing the specific characteristics of the uses in which hydrogen will be consumed. Most likely, the largest contribution will come from the decarbonization of the transport sector [19]. Within this, collective and heavy transport, because it is more than likely that light vehicles will be fully electrified. In vehicles weighing less than 3500 kg, the electric motor alone can typically meet the power needs, as the battery's storage capacity is usually enough. However, this is not the case for heavier and collective transport vehicles. Charging the batteries necessary for these vehicles would require 10 to 12 h of charging using fast chargers (in the order of 50 kW), which would force an increase in the fleet to cover the same services. Then, hydrogen is a good option since, in a time interval of less than 15 min, a fuel cell vehicle can be charged. There is the added problem of the significant increase in installed power that would be required where there are now interchanges or bus stations. Although hydrogen would also mean an increase in the electrical demand for its production, this increase could be realized anywhere on the island, depending on the availability of grid connection points. In fact, the electrolyzers would probably be installed preferably downstream of the electrical substations, where there is greater connection capacity and sufficient generation to guarantee the balance per node. The use of EVs in heavy transport also has the added disadvantage of energy density, such that lithium-ion batteries can weigh up to three times as much as hydrogen ones, which also affects mobility, being weight is directly related to vehicle power.

Hydrogen, with a high probability, will be key for the maritime sector since it is practically impossible for large ships to consider the use of batteries, as their routes usually cover long distances. Finally, another possible use of hydrogen would be in different industrial applications and in the service sector, mainly cogeneration or even trigeneration (heat-electricity, heat-heat-electricity), focusing mainly on those processes that are large consumers of heat. There is also the possibility of using hydrogen in air transport through the production of synthesis fuels, but this use is at a lesser stage of development and has not been considered a significant contribution by 2040. Finally, it could be used for re-electrification, but in the scenario presented in this work, there is an interconnection between the different islands, which means that this use is not a priority either. Assimilating the behavior of the Balearic Islands with that of the Canary Islands (Spain), more than 60% of hydrogen demand would come from road transport, around 30% from maritime transport and only 10% from the industrial sector and tourism [27]. Therefore, it is considered that the hydrogen demand curve would be practically constant throughout the day. Since the distribution of $H_2$ by trucks has been considered, it would be difficult to use pipelines to the different islands and their interconnections, given the relatively low consumption in most cases [28]. Therefore, it could only be economically viable in some cases for the islands, those with the highest consumption and at points close to a hydrogen production center. Another aspect to consider is the seasonality of $H_2$ consumption. Among the uses described, only the service sector could present seasonality, but given the climate

of the Balearic Islands, this would not be very significant. Moreover, this consumption represents around 10% of overall consumption, so it could be considered that there is no seasonality. For these reasons, $H_2$ production has been considered in this study to be uniformly distributed daily between 12 h (8:00 a.m. and 8:00 p.m.), as it has also been suggested previously in similar research studies [13]. However, to account for the inherent variability that exists in end-use processes, a random hourly variability of up to 20% has been implemented. In addition, hydrogen demand can vary from day to day due to several factors, such as weather conditions, economic factors, changes in transportation patterns, daily tourist occupancy, and so on. For this reason, it has been considered that hydrogen demand can vary by up to 10% between consecutive days [13]. It is noteworthy that the Balearic Islands experience consistent weather conditions, which mitigates the likelihood of substantial fluctuations in hydrogen demand. Furthermore, the distribution of hydrogen will take place at commercial distribution points, such as hydrogen stations, that possess storage systems allowing for adaptable management with a margin at each site.

### 3.2. Renewable Power Generation System

As mentioned above, the total decarbonization of the economy should be a reality in the European Union by the middle of the current century. In addition to the aforementioned plan, there exists an even more ambitious initiative in the non-peninsular territories of Spain aimed at leading the ecological transition and the establishment of a decarbonized energy system [19,22]. This endeavor seeks to expedite decarbonization by 10 years, requiring the development of various strategies related to crucial facets of the Balearic energy system. These include but are not limited to self-consumption, the adoption of electric vehicles, storage utilization, the integration of hydrogen in challenging electrification scenarios and the promotion of demand management strategies [18,19]. For this reason, several projects are being developed in the Balearic Islands aimed at minimizing its high dependence on fossil fuels, but they are still far from achieving the ambitious goal. Therefore, this study shows an analysis that provides information on the characteristics of a fully renewable system capable of achieving this decarbonization of the economy, additionally covering all the final energy uses through the use of electricity and hydrogen. To this end, the islands have an enormous advantage, given the abundant natural resources of the archipelago, especially the wind and the sun.

### 3.2.1. The Solar Photovoltaic System

Due to the excellent radiation characteristics of the Balearic Islands, solar resources must be an essential part of power generation in a renewable system. This solar resource can be estimated through the POWER Data Access Viewer of NASA [29]. According to their documentation, the hourly solar data are based on satellite observations together with surface solar irradiance information from NASA's Global Energy and Water Exchange Project (GEWEX)/Surface Radiation Budget (SRB) Release 3 and NASA's CERES Fast Longwave and Shortwave Radiative project (FLASHFlux).

Initially, it was thought to sample all the historical data of the database (about 20 years) in order to obtain the mean hourly values. However, this approach would attenuate the variability of the resource since we would be working with hourly mean values. For this reason, it was finally decided to use the year 2019 since, after analyzing the database, it was found to be a "typical" year in terms of solar irradiance. The measure of the atmospheric conditions, the brightness index, has also been considered so that the fraction of solar radiation transmitted through the atmosphere to the earth's surface has been taken into account in the calculations. This energy represents a potential global horizontal irradiance of 1699 ESH/year (equivalent hours of sunshine). These data are used in the current research, assuming that this irradiance is maintained until the year 2040. As for the characteristics of the possible solar PV panels used to take advantage of this solar resource, it has not been detected that there are major differences between the different models on the market, so Table 2 provides the most important information on the basic characteristics

of those used in the solar PV installation of this analysis. The extra cost and complexity of using sun tracking systems have been ruled out, so the use of conventional panels has been considered.

**Table 2.** Inputs used for the PV system [30].

| | |
|---|---|
| Lifetime (years) | 25 |
| Derating factor (%) | 90 |
| Tracking system | No tracking |
| Used panel | Vertex 550+ |
| Temperature coefficient of power (%/°C) | −0.38 |
| Peak Power (W) | 550 |
| Nominal operating cell temperature (°C) | 45 |
| Efficiency of the panel at standard conditions (%) | 21.1 |
| Cost (€/kW) | 1300 |
| Operation and Maintenance (O and M) cost (per 1 MW peak power) (€/year) | 3500 |

Finally, a maximum value of solar PV energy to be installed has also been considered, given the conditions of the islands [20]. The power to be installed in each of the islands of the archipelago is obtained from the optimal estimates of the analysis of self-consumption of a report of the government of the islands so that a value of 2.323 GW maximum power to be installed on roofs is considered. Specifically, it has been considered as a standard, for example, an occupancy of 40% for industrial roofs and extensive residential roofs, and 30% for intensive residential roofs, although extra considerations have been made depending on the age of the building, location, orientation, etc. More information is detailed in the aforementioned document of the Balearic government [20]. This document emphasizes that this capacity could be significantly increased by the installation of solar farms, reaching approximately 73 GW (this calculation does not consider land associated with natural areas, areas of the Nature network, wooded areas and other pre-protected areas).

### 3.2.2. The Wind System

Wind in the Balearic Islands is the other major resource available. The magnitude of the wind resource can be assessed using the POWER global wind data access viewer developed by NASA [29]. The meteorological information provided by this database is based on Goddard's Global Modeling and Assimilation Office (GMAO) Modern Era Retrospective-Analysis for Research and Applications (MERRA-2) and GMAO Forward Processing—Instrument Teams (FP-IT) GEOS 5.12.4 near-real-time assimilation model products. Following an analogous reasoning to that used for the solar resource, the wind data of the historical data of the aforementioned database (about 20 years) have been consulted, having discarded the use of averages. Finally, the hourly data for the year 2019 have also been selected as representative. This information is provided to the code to quantify the available wind resource in order to estimate the production of the wind turbines. As for the solar resource, it has been assumed that these wind data will be maintained until 2040.

In the case of wind, an analysis of the most suitable possible sites for the islands has been carried out so that Figure 6 shows a view of the average wind speed in the archipelago. As can be seen in Figure 6, wind turbines can be installed in various locations, both for onshore and offshore technologies. However, in this study, it has been considered that the best option was the use of offshore technology from the land occupation side. This technology is slightly more expensive, but on the other hand, it has a higher production since the average wind speeds are more stable and higher in offshore sites. All this, together with the aforementioned criterion of land occupation, has led it to be considered the best option. Table 3 shows a summary of the technical data sheet of the selected wind turbine.

As mentioned for wind, there are no major technical differences between the different commercial models available.

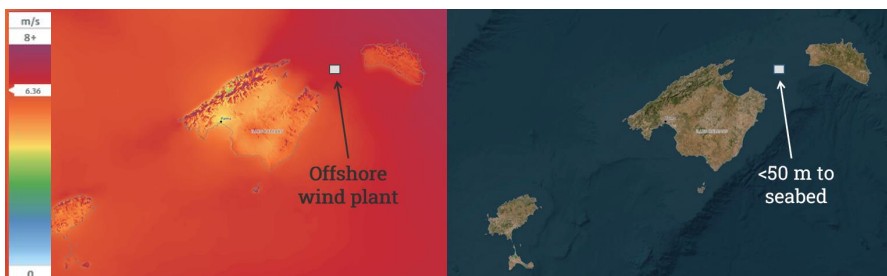

**Figure 6.** The offshore wind resource and optimum wind farm site in the Balearic Islands [29].

**Table 3.** Datasheet of the wind turbine.

| Wind Generator | Enercon E-126 |
|---|---|
| Rated power (MW) | 7.58 |
| Rotor diameter (m) | 127 |
| Height to the axe (m) | 135 m |
| Total height (m) | 197 m |
| Lifetime (years) | 25 |
| Cost of the system (M€/turbine) | 18.1 |
| M€/MW | 2.39 |
| O and M cost (M€/year) | 0.49 |

### 3.2.3. The Biomass System

Biomass energy production mainly relies on wood and its derivatives, but it can also use other organic materials from agriculture, animals, municipal waste, and human and animal excreta. Additionally, biomass can be a backup system in fully renewable systems. There are different ways to extract energy from biomass. The simplest one is direct burning, which produces heat by setting biomass on fire. Another option is gasification, which involves heating biomass with controlled amounts of oxygen and steam to produce gas and liquid fuels. There are also other methods, such as microbial conversion, pyrolysis, chemical conversion, etc., but only burning and gasification are commercially viable. Burning is more suitable for steady and continuous production, while gasification is more flexible and versatile. Burning also tends to emit more GHG gases than gasification, about 30% more [31]. Gasification systems that produce both heat and power (CHP) can achieve electrical energy conversion efficiencies of up to 35% [32]. Chowdhury's PhD thesis [33] provides a detailed analysis of different biomass gasification systems, where the author reports electrical efficiencies between 22–35% with costs ranging from 6.5 to 14 c€/kWh when gasification is combined with a gas turbine. If integrated gasification combined cycle (IGCC) technology is used, efficiencies could reach 40–50%, with electricity generation costs between 10.5 and 13.5 c€/kWh and capital costs from 1035 to 5440 M€/MWe. The Operation and Maintenance cost are usually split into fixed (which include personnel, scheduled maintenance, equipment replacement, etc.) and variable costs (which include all costs that depend on the generation except biomass fuel), the first around 2.5% of the initial investment, while the variables ones are around 3.5 €/MWh [33]. However, both technologies are not yet fully developed commercially.

Particularly, in the case of the Balearic Islands, the availability of biomass resources is relatively high if considerations are made, including the possibility of energy crops, given the scarce exploitation of agricultural land at present [20]. For the exploitation of these resources, it is possible to design small power plants with fuels from agricultural activities, although there would be problems to guarantee the supply of larger consumptions. To

solve this situation, resources should be available on a regular basis, with quality and at an acceptable cost, for which the option of energy crops should be exploited. The theoretical potential for electricity generation considering the biomass that is collected in pruning remains, forests, etc. and that is currently destined for other uses is about $1.8 \cdot 10^5$ MWh. However, if energy crops are considered, being the potential production of biomass through these energy crops on land that is currently out of cultivation, electricity generation capacity increases to approximately $1.2 \times 10^6$ MWh.

### 3.2.4. The Storage System

Storage systems are crucial for ensuring a reliable and flexible energy supply for any generation system. They are especially important for isolated regions with stand-alone grids, such as islands, where the goal is to manage the excess electric energy that occurs when a fully renewable system is implemented [34,35]. Different energy storage technologies have been developed to store this surplus electricity, such as converting electrical energy into mechanical, thermal, gravitational, electrochemical and chemical energy. However, only two storage technologies can provide the storage capacity needed for the systems discussed in this study: reversible pumped storage plants and mega-batteries.

In the Balearic Islands, there are currently no dams already built that would allow the installation of reversible pumped-storage power plants. At this point, it is worth mentioning that there is always the possibility of carrying out a project similar to the one existing in El Hierro (Canary Islands), which combines the use of the abundant wind resource with a hydraulic energy storage system [36]. However, for the implementation of this system without having the previous infrastructure, the project increases in cost and complexity, so probably the option of using mega-batteries becomes the priority option. Additionally, in the Deloitte monitor report, it is mentioned that batteries in the medium to long term will probably be a very suitable option [19].

An advantage of batteries is that this technology can store electrical energy without converting it into other energy intermediate forms. This technology is becoming more popular mainly because of the increasing demand for EVs and the need to store the excess electricity generated by renewable energies. This has led to major improvements and cost reductions in electric battery technology. However, there is still room for enhancing their lifespan and promoting their reuse and recycling. However, the current trend of continuous improvement is expected to persist in the future. As mentioned before, the only feasible option today is to use mega-batteries. They would work in the same way as pumping stations, storing excess energy and releasing it when needed. Table 4 shows the specifications of the mega-batteries system used, Gildemeister 250 kW-8 h Cellcube, batteries which use the technology of vanadium redox flow, which provides a much better relationship between the storage capacity to the peak power.

**Table 4.** Standard system specifications of the selected battery system [37].

| Battery | Gildemeister |
|---|---|
| Maximum AC power (kW) | 250 |
| Energy Available (MWh) | 2.480 |
| Round-Trip System Efficiency | 70% |
| Cost of the module (€) | 460,000 |
| O and M cost (€/year) | 5300 |
| Lifetime (years) | 25 |

## 4. Results

This section presents and discusses the main findings of the proposed scenario. The economic and technical aspects will be taken into account, analyzed and discussed. The system has been optimized to cover the demand entirely, both for electrical and hydrogen.

To achieve this 100% demand coverage, the different available resources and generation systems, namely wind, sun and biomass, and their conditioning factors previously mentioned (maximum available capacities of solar PV and biomass mainly because of land occupation criteria) have been considered. Due to the unavoidable need to use storage technologies to manage a fully renewable system of this scale, the implementation of a significant mega-battery storage system has been needed. In other words, the power to be installed for wind, solar PV and biomass has been determined, along with the power and storage capacities of the mega-batteries and the Spanish mainland grid connection to make the system self-sufficient. Added to the dimensioning of the hydrogen production sub-system.

*4.1. Technical Characteristics of the Systems*

Tables 5–7 summarize the main characteristics of the power generation, storage and hydrogen production systems proposed for the Balearic Islands. A significant aspect is that these islands have an interconnection with the mainland, which gives them greater flexibility in the implementation of fully renewable systems. As shown in Table 5, only sources with very low GHG emissions have been used (wind, solar PV and biomass), together with exchanges with the grid. By far the largest installed power is solar PV, with more than 75% of the total (six GW of nominal capacity), followed by wind with around 15% (slightly below 1.2 GW) and lastly, gasification technology with around 7.7% (600 MW). The capacity factors of solar PV and wind power sources are high, with values quite similar to those typical of existing installations in mainland Spain, around 20% and 40%, respectively. Noteworthy is the importance of biomass, which, together with the grid and batteries, provides the necessary flexibility to the system to take advantage of the large installed capacity of solar and wind generation. This highlights the relatively low percentage of energy waste (below 9%) for a fully renewable system, especially considering that not only are all end uses of energy being covered by a GHG-free system but also by electricity generation combined with hydrogen production.

Table 6 summarizes the major characteristics of the storage system. It has been decided to only use battery storage since the islands do not have the pumping sites required to use this system, although it is true that the necessary infrastructures could be built and this resource could be used. Nevertheless, in this case, the pumping station installations would require high initial investments; it would be necessary to build from the dams, the hydraulic circuit to the power line and the rest of the installations, probably even a desalination plant to provide the necessary water for its exploitation. All this, together with the more than likely opposition to the large landscape interventions required by the population, means that given the flexibility provided by the network, together with the installed capacity of biomass and the mega-batteries, it has not been necessary to resort to the extra storage of reversible pumping. As shown in Table 6, the total installed capacity of this technology has been about 2.64 GW with a total energy storage capacity of slightly over 21 GWh; a total of 8700 units have been installed. The system has been able to recover more than 10% of the electricity generated, which otherwise would have been impossible to use. In other words, this amount was not wasted but fed back into the grid when needed, albeit with a round-trip efficiency of 70%.

Table 7 shows the main characteristics of the hydrogen production sub-system. From the results shown in the table in Figure 7, it can be noted that the total installed power of the electrolyzers is less than 2 GW, which has a relatively low capacity factor (19.2%). This is due to the fact that $H_2$ production takes place when there is excess electricity generation, i.e., given the high weight of solar generation, production takes place especially in the central hours of the day. So from the calculations performed, it has been shown that it is more convenient to oversize the $H_2$ production sub-system than any of the sub-systems of electricity generation and/or storage. Another aspect to consider, which has not been reflected in the table, is that the water electrolysis process produces a significant amount of oxygen as a by-product (approximately 9 kg of $O_2$ for each kg of $H_2$ produced), so

this oxygen could be used in different processes. In this case, there would be more than $4.5 \times 10^5$ tons of $O_2$. Some possible applications would be, among others: sanitary, welding, oxy-cutting, and replacement of air in wastewater aeration. Therefore, it is an extra benefit obtained from this by-product that could be considered, although this is beyond the scope of this work since it would be necessary to consider the cost associated with this infrastructure (approval, storage, distribution, etc.).

**Table 5.** Summary of the electric generation systems and grid balance.

| Generation Systems | Solar PV | Wind | Gasifier | Grid Purchases | Grid Sales |
|---|---|---|---|---|---|
| Rated Power (MW) | 6000 | 1167 | 600 | 650 | 650 |
| Mean Output (MW) | 1165 | 372.7 | 571.8 | 221.5 | 88.9 |
| Hours of operation (h/year) | 4048 | 8241 | 2047 | 4151 | 1396 |
| Mean Energy Output (GWh/day) | 27.96 | 8.94 | 3.21 | 5.32 | 2.13 |
| Capacity Factor (%) | 19.40 | 31.90 | 22.30 | 34.07 | 13.68 |
| Equivalent Hours (hours/year) | 1699 | 2794 | 1953 | 2985 | 1198 |
| Total Generation (GWh/year) | 10,205.4 | 3264.5 | 1170.5 | 1940.2 | 778.8 |
| | 61.6% | 19.7% | 7.1% | 11.7% | 4.70% |
| | | | 16,581 | | 779 |
| System Electric Surpluses (%) | | 8.93 | | | ---- |

**Table 6.** Summary of the electric storage system.

| | |
|---|---|
| Rated Power (MW) | 2624 |
| Rated Capacity (GWh) | 21.05 |
| Round Trip Efficiency (%) | 70 |
| Electric Energy Input (GWh/year) | 2856 |
| Electric Energy Re-Feed (GWh/year) | 2000 |
| Electric Energy Losses (GWh/year) | 857 |

**Table 7.** Summary of the hydrogen production system.

| | |
|---|---|
| Rated Power (MW) | 1700 |
| Mean Output (MW) | 325.73 |
| Mean Energy Output (GWh/day) | 3.84 |
| Hours of operation (h/year) | 4303 |
| Capacity Factor (%) | 19.2% |
| Equivalent Hours (h/year) | 1682 |
| Total $H_2$ Production (tons/year) | 51,000 |

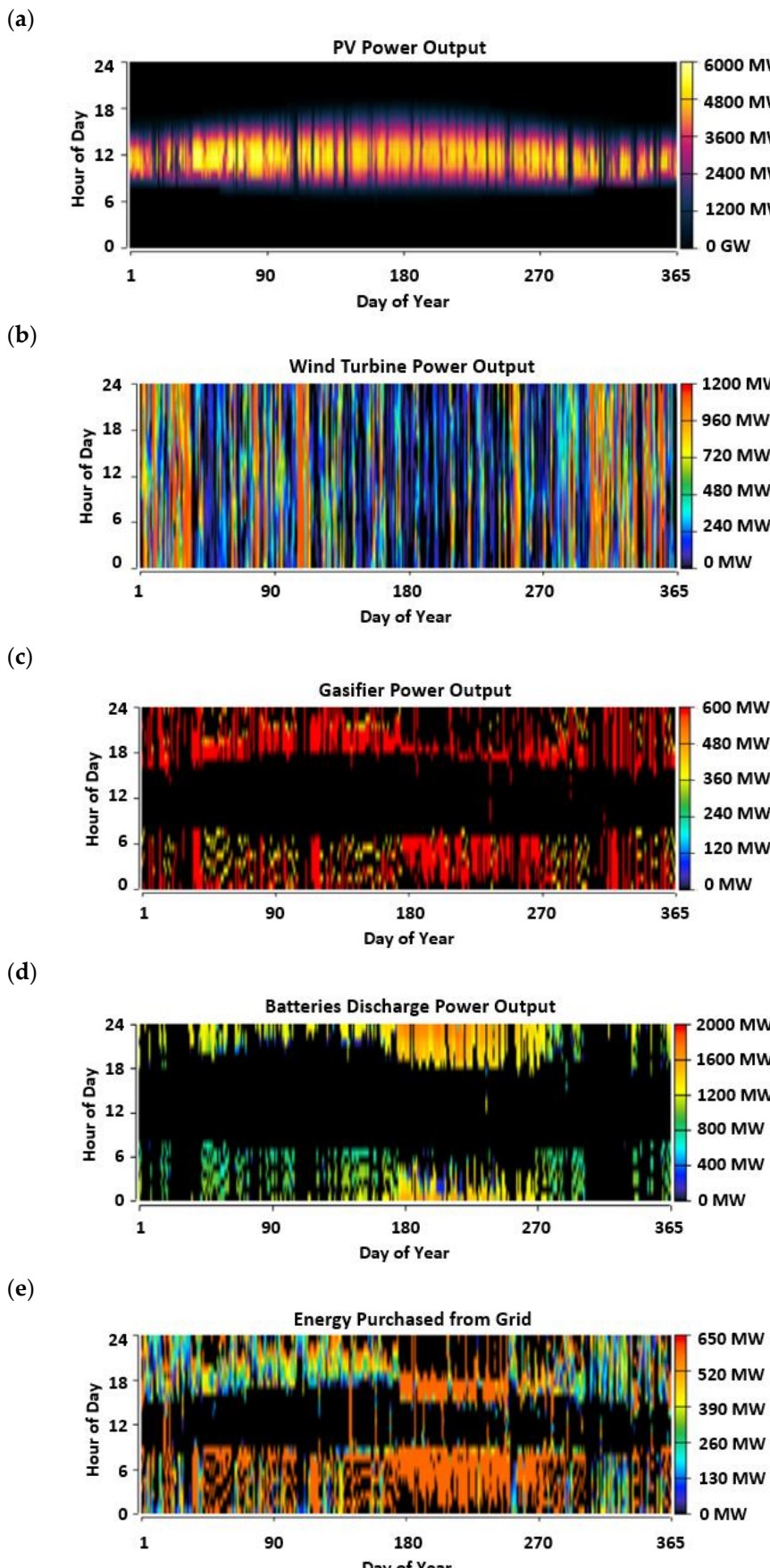

**Figure 7.** Yearly performance of the energy sources. (**a**) Solar PV; (**b**) Wind; (**c**) Biomass; (**d**) Batteries discharge; (**e**) Grid Purchases.

To provide an overview of the overall performance of the proposed system, Figures 7–12 show the behavior of the different generation sources and the two consumptions. The first two provide the general view of the performance of the generation sources (in fact, generation, re-feed of the storage system and grid purchases) and the electric consumptions (charge of batteries, sales to the grid and electrolyzers consumption). Figure 9 displays the monthly state of charge of the batteries. While Figure 11 shows the performance of the entire system for four typical days of the year. Finally, Figures 10 and 12 focus on the behavior of the hydrogen production sub-system, showing the monthly evolution of the hydrogen reservoir inventory, while the last figure displays the performance of the hydrogen sub-system for typical winter and summer days.

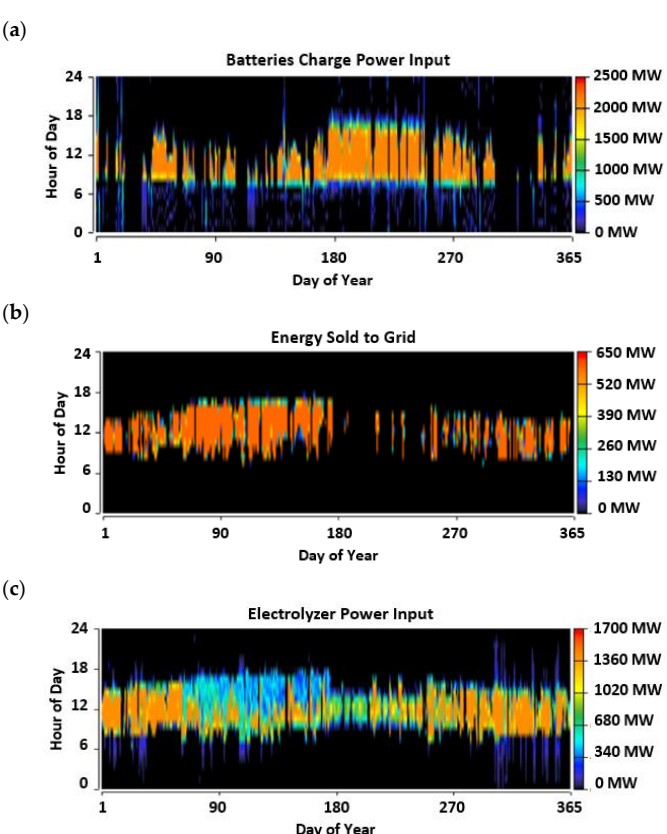

**Figure 8.** Yearly performance of the electric consumptions. (**a**) Batterie charge; (**b**) Grid Sales; (**c**) Electrolyzer.

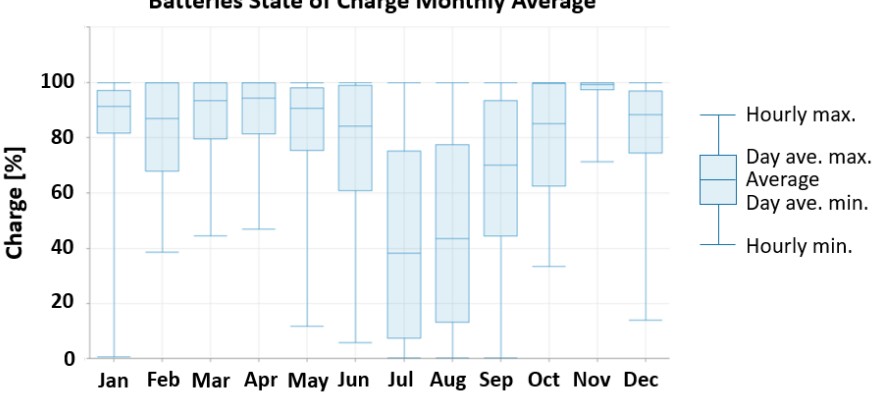

**Figure 9.** Monthly evolution of the charge level of the mega-batteries storage system.

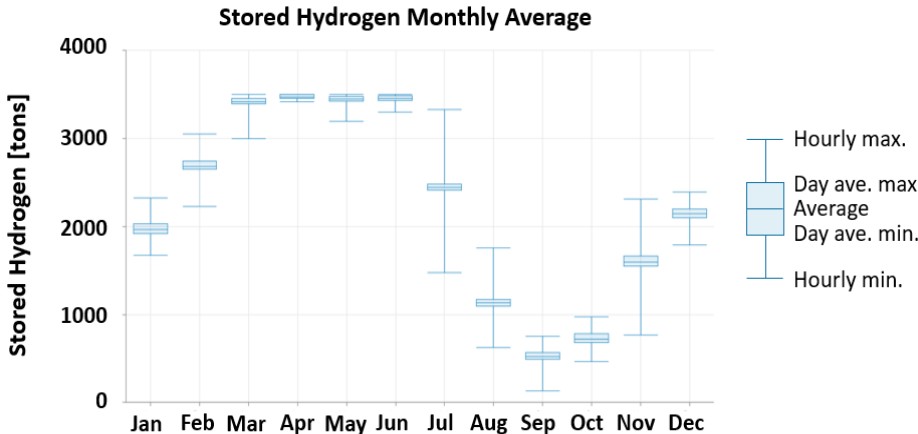

**Figure 10.** Monthly evolution of the hydrogen storage tank.

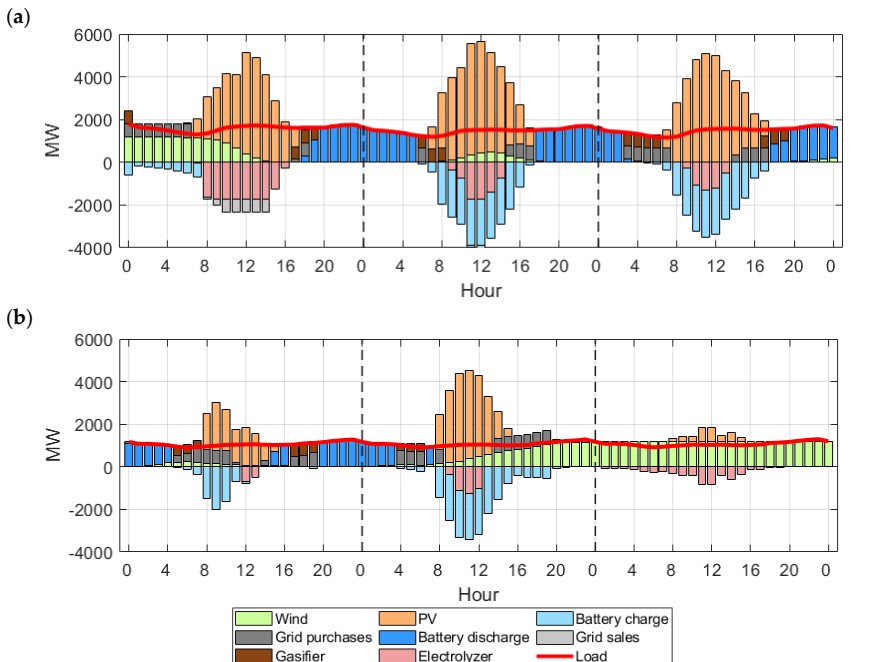

**Figure 11.** Hourly performance of the proposed systems in a typical three-day series of (**a**) summer and (**b**) winter.

Particularly, Figures 7 and 8 show the annual representation of the behavior of the generation and consumption components. Figure 7 shows solar photovoltaic, wind and biomass electricity generation, also including the contributions of both the general grid and the electricity refeed from the mega-batteries. As shown in the first figure, the solar resource has its maximum contribution during the summer months (Figure 7a); between the months of April to September approximately, there is greater insolation and more daylight hours, therefore, the power and total energy generated are higher. As for wind power generation, as shown in Figure 7b, there is very high variability and unpredictability, although the highest contributions occur during the winter months, while the months between June and September (days 150–280) have the lowest values, but in any case, there are many time periods in which the wind generation is zero or almost zero. Then, the combination of wind and sun profiles makes nights the critical time period for the overall performance of the system. On the one hand, solar generation (more than 60% of the energy generated) is obviously null during the night, which means that other sub-systems must cover this deficiency (i.e., wind, biomass, batteries and/or grid). In addition, as mentioned above, wind generation (the second most important source of generation, accounting for

almost 20% of total generation) is highly unpredictable, with long periods of wind shortage, with a greater degree of severity in the summer months. All this means that during most nights, especially in the summer months, it is necessary to have a very strong backup from other sources to solve this problem. Figure 7d shows the battery performance diagram, where it can be seen how, especially in the summer months, the batteries are the main source of supply for the demand during the first hours of the night, reaching their complete emptying during most of the time after midnight approximately; Figure 9 shows how their charge reaches zero during those summer months. Therefore, as shown in Figure 7c,e, it is at these times when the biomass gasifiers and purchases from the peninsular grid are most important. In fact, in the summer months (mainly between days 180 to 270), from approximately midnight onwards, both are at their maximum power until 7 a.m. (solar time), which is when solar generation begins to make an important contribution, given that it has already been dawn for a couple of hours.

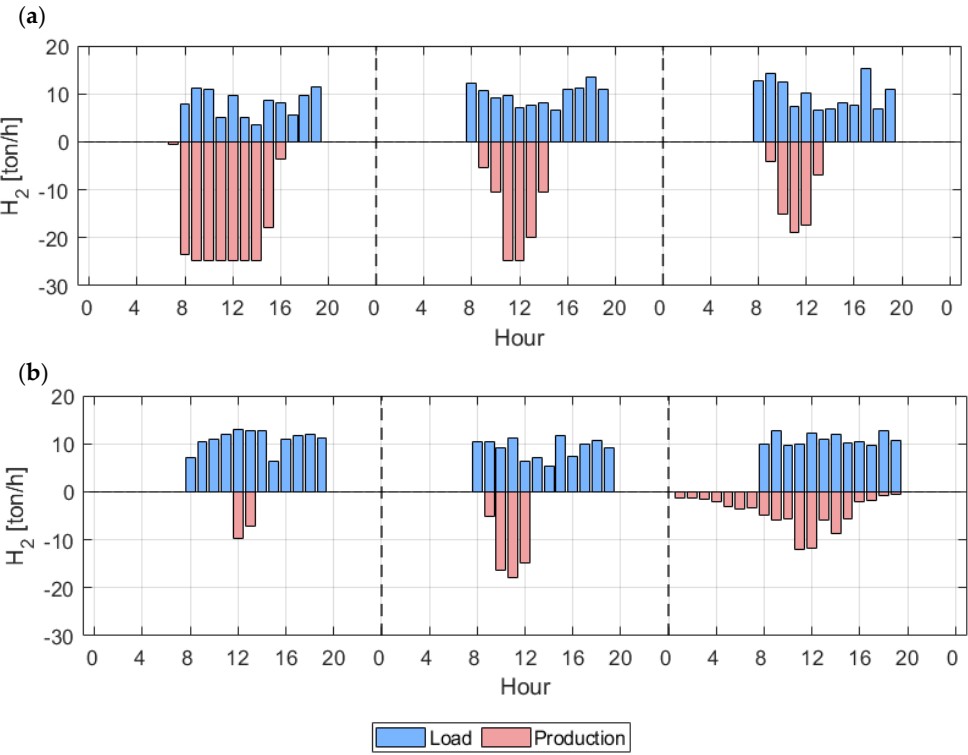

**Figure 12.** Hourly performance of the hydrogen production systems in a typical three-day series of (**a**) summer and (**b**) winter.

As for the recharge of the battery system (Figure 8a), it should be noted that, in addition to what happened in its discharge (Figure 7d, where it was discharged in the early hours of the night), this occurs mainly in the central hours of the day (produced with excess energy, especially from the sun). Additionally, as the batteries become essential in the summer months (Figure 7d), it is then when during the night they suffer deep discharges, and during the day they are recharged (Figure 8a) so that on many occasions, they are not fully charged (as shown in Figure 9). This situation means that on most days, it is necessary to resort to other systems in a very significant way (biomass and purchase from the peninsular grid).

Figure 8a represents the charge power input, while Figure 8b shows the sales to the peninsular grid, where it can be seen that in the summer months, practically no energy is sold at all. While in the rest of the year, there are significant sales, especially during the spring, when solar and wind generation are very important. It should also be noted that purchases from the grid are made throughout the year (Figure 7e) but become more significant in the summer months when the system presents its most critical performance

intervals (purchase during long periods of time of the maximum 650 MW). Thus, the final balance with the peninsular grid shows a deficit value, with purchases amounting to 11.7% of total energy consumed while sales amount to slightly less than 5%.

Analyzing the performance of the electrolyzers, as shown in Figure 8c, sway that it is also during the summer months when the critical performance of this sub-system is the highest. So that during these summer months, the hydrogen storage tank is progressively emptied (Figure 10); as shown in Figure 8c, the power and time of use of the electrolyzers are much lower than in the winter months. So the hydrogen tank begins its progressive emptying in April, reaching its minimum level in September; it is filled again progressively until March, and it remains practically full until July, when it begins to empty again. As shown in the figure, only the summer months will be critical since there is not enough energy available for the production of the hydrogen demanded, and the excess stored during the rest of the year must be used.

Regarding Figures 9 and 10, it should be noted that the behavior of the charging and discharging cycles of the batteries and the filling of the hydrogen tanks show many similarities. As the system, in general, has its critical performance fundamentally at the end of the summer months, it is then when both systems present their minimum values. While from the beginning of autumn, both systems begin to present increasing values of battery charging and hydrogen tank filling until mid to late winter, when they reach their maximum values. They remain at these values until late summer, when the progressive discharge or emptying begins again, respectively, so that a new annual cycle of similar characteristics would be carried out again.

To analyze the performance of the system as a whole in a more detailed way, Figure 11 shows the hourly values for three typical consecutive days for both summer and winter months. As can be seen in this figure, the average daily electricity demand in summer is higher than in winter. However, in spite of this, for the central hours of typical summer days (Figure 11a), due to the higher contribution of solar PV (due to higher average irradiation and more hours of sunshine and the high installed capacity, 6 GW), the system does not present problems to provide the energy required by the loads. However, the difficulties in covering the supply start in the case of low or no wind (quite the usual case in summer, as shown in Figure 7a). In this case, the system must resort to the use of the batteries practically at their maximum power, as well as to the use of the gasification system and the purchase from the grid. In fact, in the last two nights, the batteries are exhausted, the first one around 5:30 in the morning and the second one shortly after 1:00 a.m., having to resort in both cases to the use of the gasifiers and purchases from the peninsular grid, both at their maximum power (600 and 650 MW, respectively). While for typical winter days (Figure 11b), as long as there is a significant contribution from wind generation, the system does not need to resort to any of the other systems (storage, gasification, grid purchases), while in the absence of wind, the system must resort to these systems, but in this case, to a lesser extent due to the lower load requirements and the fact that the battery systems are in a higher degree of load (Figure 9). Thus, the high power and capacity of the batteries mean that the system does not present major problems, although it does resort to the use of biomass and purchase from the grid, but to a much lesser extent than in the case of typical summer days.

Similarly, for a more detailed analysis of the hydrogen production system, Figure 12 shows the behavior of this hydrogen production and consumption system during three consecutive "typical" summer and winter days. These days correspond to those shown in the behavior of the system (Figure 11). The graphs in the previous figure show a constant $H_2$ demand but with some variability (a random hourly variability of up to 20% and up to 10% between consecutive days). On typical winter days (Figure 12b), as the $H_2$ reservoirs are practically full (Figure 10), the system produces at maximum capacity for several hours so that when the reservoir is completely full again, the system only has to cover approximately current $H_2$ demand, leaving hydrogen production until the next day. However, on a typical winter day (Figure 12a), the system works at its maximum power

during the central hours of the day, while surplus electricity is produced (the first day when there is wind). However, from the second day on, there are practically no surpluses available, so the system is not able to produce the hydrogen needed to cover the demand for it, so it has to resort to part of the reserves, emptying the storage tank progressively throughout these months (as discussed above and shown in Figure 10).

It should be noted that, contrary to what one might intuitively think, the late summer months are the most critical for the system. As mentioned above and corroborated in the figures analyzed, all months except the summer months (especially the spring months) have an electricity generation and hydrogen production capacity higher than the aggregate demands. Thus, except in the summer months, when the system has to resort to the maximum use of biomass gasification generation, the storage system and the purchase from the mainland grid, in the rest of the year, the system has no major difficulties in covering the electricity demand, not having to resort to these systems on a continuous basis. This need is mainly due to the existence of long periods of time with little or no wind resources.

*4.2. Economic Analysis*

Table 8 presents the main variables associated with a financial analysis of the proposed system. It is important to note that the total costs are directly influenced by the installed capacity of each system, thus establishing a direct relationship with the information presented in Table 5, which outlines the electrical generation and power of each system. The total capital cost necessary to carry out the installation of the entire electric and hydrogen system is 27,306 million euro. Of which the initial investment accounts for almost 75%, while the Operation and Maintenance costs of the different sub-systems account for about 23%, while finally, the replacement costs account for less than 2% since only the replacement of the hydrogen system is foreseen during the life of the project, as the electrolyzers used have about 100,000 h of operation.

**Table 8.** Capital, replacement, O and M, and total discounted costs for the sub-systems of the scenario.

| Sub-Systems | Capital (M€) | Replacement (M€) | O and M (M€) * | Total (M€) * |
|---|---|---|---|---|
| PV | 7800 | 0 | 331 | 8131 |
| Wind | 2790 | 0 | 1189 | 3979 |
| Gasifier | 1943 | 0 | 2308 | 4250 |
| Grid | 359 | 0 | 1006 | 1365 |
| Mega-batteries | 3910 | 0 | 710 | 4620 |
| Interconnection | 1613 | 0 | 242 | 1855 |
| Hydrogen | 2054 | 458 | 1335 | 3848 |
| TOTAL | 20,468 | 458 | 7120 | 28,048 |

* Total costs over the 25-year life of the project.

Breaking down the total costs by sub-systems, the cost of solar photovoltaic generation is almost 30%, followed by the cost of mega-batteries with almost 17%, while the wind and hydrogen sub-systems both have around 14%, being slightly higher than the wind sub-system, followed by the gasification sub-system with almost 13%. The two sub-systems related to electricity transmission, i.e., interconnections and upgrading of the internal transmission grid, with almost 7% and 5%, respectively, occupy the two lowest positions. As for the capital costs, which is the contribution with the highest weight by far, the order of importance in the contributions is similar, although slightly changing the weights; in this case, the solar photovoltaic system is the largest contributor with around 39%, the mega-batteries also increase their weight with almost 20%, the wind sub-system has a weight of almost 14%, while hydrogen, gasifier and interconnection have a cost of around 10%, relegating to the last position the network upgrades almost 2%. The Operation and Maintenance costs present some remarkable aspects; the solar photovoltaic sub-system

and interconnections present very low costs, while at the other extreme is the gasification sub-system with almost 25%, close to the wind generation sub-system, with almost 20%, followed by the internal distribution network and mega-batteries with almost 15% and just over 11%, respectively.

In the analysis of the relationship between capital costs versus operating costs over the lifetime of the system, there are some situations to highlight. The costs of the solar sub-system come almost exclusively from its installation (more than 95%); something similar occurs in the interconnections and mega-batteries (with about 85%), wind also presents a high value (about 70%), while the hydrogen production and biomass gasification sub-systems present proportions of approximately 50% in capital and Operation and Maintenance costs during the useful life. Finally, the Operation and Maintenance of the internal network of the islands has a weight much higher than the initial investment (almost 75%–25%).

The hydrogen system analyzed in this study considers the different costs from production to distribution at the final consumption points, Table 9. Thus, the system consists of several components, and the referenced table shows a breakdown of the different contributions involved in hydrogen production. The largest contribution comes from the electrolyzers, accounting for about 40% of the total system costs, followed by the costs of the storage system and the compression it undergoes, with about 28%. The following contributions are those of the second compression and transportation, both with similar contributions of 15% and 13%, respectively. Finally, about 3% corresponds to the water demand of the electrolyzers; this water comes from seawater desalination, given that the island is deficient in water resources.

**Table 9.** Summary of the hydrogen production costs for the project lifetime.

| | |
|---|---|
| Electrolyzer (M€) | 1554 |
| Desalination plant (M€) | 120 |
| $H_2$ tank and 1st compression (M€) | 1082 |
| Second compression (M€) | 582.2 |
| Transport (M€) | 510 |
| Total Costs (M€) | 3848 |
| Total $H_2$ production (tons $H_2$/year) | 51,000 |
| LCOH (€/kg $H_2$) | 4.79 |

Table 10 summarizes the LCOE costs of the different technologies; these values, together with their contribution in percentage, provide the final average LCOE of the system. The average LCOE value is 13.75 c€/kWh, which is relatively low for a system of these characteristics since it is a system with a connection to a central grid but only capable of covering around 34% of the system's peak power (peak demand power of 1915 MW compared to the 650 MW of maximum power of the interconnection with the mainland). In addition, the system covers all energy end-uses, i.e., electricity generation, and part of this is used for hydrogen production. As shown in the table, all generation systems have assumable LCOEs, being the exchanges with the mainland a very important contribution to providing the mandatory system stability of these semi-isolated regions of the grid. Solar and wind generation present very competitive values, but due to their variability and unpredictability, they introduce an extra need for reliable systems. Therefore, the system requires, in addition to the aforementioned continental and inter-island interconnections, a large storage system; in this case, a mega-batteries sub-system has been selected (Table 6). In addition, there is quite a high biomass power (acting as a base generation system or backup), so it is available for any contingency. Although the cost of this system is high due, on the one hand, to its occasional use and, on the other hand, to the intrinsic difficulties existing in the islands for the exploitation of these resources, but is offset by the extra

reliability it brings to the system. Finally, the hydrogen system also has a high cost, but it should be remembered that this system provides an energy vector in those energy end uses where electricity is not a good option. Therefore, despite its high cost, it is considered the most suitable option to cover these energy uses. The use of hydrogen as an energy vector for non-electrification uses is probably the most realistic option for the future (Deloitte Monitor [19]). This report points out that decarbonizing these non-electrifiable uses would hardly imply a cost similar to decarbonizing the rest of the previous uses.

**Table 10.** Summary of the LCOE analysis of the system.

| System | LCOE [1] (€/MWh) | Weight [2] (%) |
|---|---|---|
| Solar PV | 50.6 | 61.6 |
| Wind | 77.4 | 19.7 |
| Biomass | 230.6 | 7.06 |
| Grid [3] | 44.7 | 11.7 |
| Total Generation | --- | 100 |
| Mega-batteries | 146.7 | 12.1 |
| Hydrogen | 143.7 | 17.2 |
| System Average [4] | 137.5 | |

[1] In the case of hydrogen, the LCOE is calculated from the LCOH, considering the LHV (Lower Heating Value) of hydrogen (33.333 kWh/kgH$_2$). [2] All weights are normalized against the total energy generated. [3] Grid LCOE is obtained considering the capital and Operation and Maintenance costs and the balance contribution of purchases and sales. [4] The cost of the interconnections and all the distribution sub-systems has been taken into consideration for the LCOE of the final system [24].

To conclude this economic analysis, Table 11 shows the main economic data and indicators of the proposed system. A reference value of 150 €/MWh for the grid electricity cost has been chosen, this value comes from typical values for the Spanish Islands during recent years, but higher prices are expected in the coming years; in fact, last year, the electricity costs in the Balearic Islands exceeded 200 €/MWh many times [25]. Consequently, this price range has been considered to make two forecasts of the main economic variables related to the profitability of the system. It is important to highlight that despite the strong investments of the implemented systems, the return on investment periods and the payback are relatively low. Therefore, the selected system is considered to be feasible and profitable, obviously much better in the case of high electricity costs.

**Table 11.** Summary of the economic analysis.

| | Electricity Cost for Reference Scenario | |
|---|---|---|
| | **Low** | **High** |
| Total NPC (M€) | 27.22 | |
| Operating cost (M€) | 428.72 | |
| Present worth (M€) | 3068 | 12,665 |
| Annual worth (M€/year) | 194.8 | 804 |
| Return on investment (%) | 3.4 | 6.3 |
| Internal rate of return (%) | 5.3 | 9.1 |
| Simple payback (years) | 13.7 | 9.7 |
| Discounted payback (years) | 20.6 | 12.5 |

## 5. Discussion and Main Findings of the Balearic Islands' Electricity System

The EU and, consequently, Spain are focusing great efforts on the reduction of GHG, with the ambitious goal of achieving the complete decarbonization of the European econ-

omy by 2050. To work in this direction, there are programs that encourage and accelerate this green energy transition; in particular, in the Balearic Islands, there is the Energy Transition and Climate Change Plan of the Balearic Islands [21], which presents the planning to achieve the complete decarbonization of the economy of the Balearic Archipelago, which is planned even ten years ahead of the deadline established for the rest of the State [19], being these documents the basis of the projections for 2040 made in this study.

It should be noted once again that the total decarbonization of the economy in itself presents numerous and complicated challenges. However, in the case of the islands, these are even greater, mainly due to the additional difficulties linked to their isolation, even though in the case of the Balearic Islands, this isolation is partial since, as described above, they have an important electrical connection with the mainland. However, on the other hand, islands also tend to have other advantages, such as the opportunities offered by the abundant natural resources that are usually present. In this sense, this study makes estimates for the archipelago with the total coverage of all final energy uses through electricity, reserving the use of hydrogen for those where electricity presents serious inconveniences. Following the calculations made with the software and the subsequent analysis of the different options, the main findings on the Balearic system and on the proposed scenario are listed below:

- The proposed scenario analyzes a system capable of covering all energy end uses through a generation mix formed by different sub-systems, all of them free of GHG emissions during their operation, using only renewable generation sources. The proposed mix uses electric power for those energy end uses that are "electrifiable". It uses hydrogen for those uses where electricity presents major drawbacks. Additionally, the size of the developed system is of considerable size, with more than 10 TWh per year of electric demand and almost 2 TWh per year of hydrogen production (more than $5 \times 10^5$ tons of $H_2$). Thus demonstrating the capabilities of these large-scale hybrid renewable systems for use in the future decarbonization of the economy, which combine different sources of renewable generation, storage and electricity/hydrogen as energy carriers;

- The LCOE of the resulting energy mix is relatively low for a system of the characteristics presented, below 140 €/MWh. This makes it a cost-competitive system with a reduced payback period for this type of system (a simple payback period of about 6 years in the scenario with the high electricity costs forecast);

- The existence of a hydrogen production sub-system has increased the flexibility of the system. On the one hand, it has contributed to the decarbonization of approximately the last 10% of energy end-uses, which are difficult to electrify (they would require an investment comparable to the electrification of the previous 90%). Moreover, on the other hand, the high power output of the electrolyzers (less than 2 GW) has enabled the system to harness a large amount of energy that would otherwise be wasted, especially in the central hours of the day (the $H_2$ production system consumes about 20% of the energy generated). However, this has been at the cost of undersizing the hydrogen production system and consequently having a low sub-system capacity factor (19%). Thus, system overruns have been low for this type of system, less than 9%;

- The high weight of solar photovoltaic generation (6 GW capacity and more than 60% of electricity generation) is due to the low LCOE of this technology. In addition, self-consumption has been exploited to its maximum level so that the installation of solar farms is minimized, thereby reducing the environmental impact caused by the large amount of land required, a very serious effect on islands that are eminently touristy. However, this great weight of solar generation entails risks, such as the over-dimensioning of the electrical system and the subsequent increase in its excesses. Given the variability and unpredictability intrinsic to generation, it is necessary to resort to other support systems that are capable of providing the necessary stability to the system;

- Along the same lines is wind power generation, which also has a considerable weight of almost 1.2 GW. With this technology, there are even greater problems associated with its intrinsic variability and unpredictability. In fact, there are long periods of time in which there is no or very little wind, which accentuates the need for other systems that provide the required reliability;
- In order to provide this stability in the proposed mix, there is a strong presence of biomass, with a power of 600 MW. This technology has a high LCOE value, above 20 c€/kWh, but plays a very important role in system stability. Since gasifiers are able to produce the energy demanded when it is required, acting as a backup system in case of any unforeseen situation;
- The high-power interconnection with the Spanish mainland (650 MW) helps to provide the necessary stability required by the system. The interconnections of the sub-systems of the four islands also provide extra flexibility to the system. So that instead of four independent systems (one for each island) isolated from a central grid, there is a single system of greater entity and also with a connection of a certain power to a central grid. In addition, these interconnections make it possible to optimize the use of the high generation and storage capacities existing on the islands. Thus, at certain times, some islands supply energy to others and/or are capable of absorbing the surpluses of other islands to a greater extent (improving the performance of the overall storage system by having a single interconnected high-capacity system), just as the connection with the mainland is capable of absorbing part of the surpluses and providing energy at times of need;
- The high power and storage capacity of the mega-battery sub-system (more than 2.5 GW of power and 21 GWh of storage capacity) has contributed greatly to the flexibility of the system, being able to absorb excess energy in times of overgeneration (more than 12% of the total energy generated has been re-injected into the grid). So it has absorbed part of the excess energy and has returned it to the grid at times when it was necessary, albeit at the cost of losing a percentage (round-trip efficiency of 70%);
- One last comment to note about the performance of the proposed mix, contrary to what one might intuitively think, the summer months are when the system is close to its limit rather than the winter months. This situation is caused by the low wind production during these months, with prolonged periods of wind shortage (almost 1.2 GW of rated power). This means that the contribution of the solar system (obviously during sunshine hours) is not sufficient to provide the daily energy demand, a demand that is, in fact, slightly higher than in the winter months. Consequently, the system has to resort to maximizing energy purchases from the peninsular grid, running the biomass generation at full load for long periods and making massive use of battery capacity;
- To conclude, we would like to comment that possible future work could focus on studies related to uncertainty analysis. These studies could focus on aspects related to the sensitivity of the system to the evolution of the future costs of the different sub-systems (generation, storage, transmission, etc.). On the other hand, an analysis of the uncertainty associated with the inevitable variability and unpredictability associated with solar and wind resources, which could significantly affect system performance, would be desirable. The different sources of demand uncertainty are also a field to be analyzed in detail (degree of implementation of demand management measures, GDP growth, population growth, degree of electrification, etc.), as they can strongly affect the determination of the optimal generation mix. This uncertainty analysis can be extended to electricity and hydrogen demand, if applicable;
- Related to the analyses associated with the sources of demand variation, we could study the effect of possible demand management policies or even the promotion of distributed generation, even at the consumer level, on the required generation mix. Specifically, the effect of measures that encourage energy efficiency could be studied, as well as those that try to shift the demand curve towards the generation

curve (prioritizing consumption at specific times) or support measures that favor self-consumption, among other possible options.

## 6. Conclusions

The proposed power generation, storage, and hydrogen production systems for the Balearic Islands demonstrate a feasible scenario using zero GHG emission sources, such as wind, solar photovoltaic, biomass, and exchanges with the Spanish mainland grid. Solar PV represents the largest installed capacity for renewable sources, followed by wind farms. The system incorporates biomass, grid, and batteries to provide flexibility and optimize the utilization of solar and wind generation. Notably, the system achieves a relatively low percentage of energy waste for a fully renewable system, indicating efficient coverage of end energy uses through GHG-free electricity generation and hydrogen production.

The storage system primarily relies on battery storage due to unsuitable hydro-pumping sites. A battery storage capacity of approximately 2.64 GW is needed, enabling the recovery of more than 10% of the generated electricity that would have otherwise gone unused. However, the round-trip efficiency of the battery system is 70%, losing about 850 GWh of energy per year. The hydrogen production sub-system utilizes electrolyzers with a total installed power of 1.7 GW. The low capacity factor of the electrolyzers is due to hydrogen production occurring during periods of excess electricity generation, primarily during the central hours of the day when solar generation is high. Oversizing the hydrogen production sub-system proves advantageous compared to oversizing the electricity generation or storage sub-systems.

The performance analysis of the proposed system reveals seasonal variations and challenges. During summer months, solar generation contributes the most, but wind generation exhibits very high variability. Nights become critical due to the absence of solar generation and the intermittent nature of wind generation. The batteries play a crucial role in supplying energy during the first hours of the night, with biomass gasifiers and grid purchases supporting the system. In contrast, winter months rely on wind generation, minimizing the need for additional systems, although showing unfavorable days of low solar and wind generation. The hydrogen production sub-system experiences critical performance during summer months, requiring the use of previously stored hydrogen due to insufficient energy availability for hydrogen production.

From an economic perspective, the total capital cost for implementing the entire electric and hydrogen system is estimated at 27,306 million euro, with the initial investment accounting for the majority. The operation and maintenance costs account for approximately 23% of the total, while the replacement costs remain relatively low during the 25-year projected duration—after which specific significant components of the renewable system may require updates. Solar photovoltaic generation represents the largest cost component, followed by mega-batteries, wind and hydrogen sub-systems, and biomass gasification. The overall economic analysis emphasizes the substantial investment required for the system's implementation. The proposed system showcases, therefore, the potential for achieving a renewable energy supply with zero GHG emissions, including the main challenges and costs for the Balearic Islands.

**Author Contributions:** Conceptualization, C.B.-E., Y.R. and D.B.; methodology, C.B.-E., Y.R. and P.B.-M.; software, Y.R. and D.B.; validation, C.B.-E., D.B. and P.B.-M.; formal analysis, C.B.-E. and Y.R.; investigation, D.B., C.B.-E., P.B.-M. and Y.R.; resources, Y.R. and D.B.; data curation, Y.R. and D.B.; writing—original draft preparation, C.B.-E.; writing—review and editing, C.B.-E., Y.R., D.B. and P.B.-M.; visualization, Y.R. and D.B.; supervision, C.B.-E. and P.B.-M.; project administration, P.B.-M.; funding acquisition, P.B.-M. All authors have read and agreed to the published version of the manuscript.

**Funding:** This research received no external funding.

**Data Availability Statement:** Not applicable.

**Acknowledgments:** The authors would like to extend their gratitude to the Ministerio de Economía, Industria y Competitividad and Agencia Nacional de Investigación under the FPI grant BES-2017-080031.

**Conflicts of Interest:** The authors declare no conflict of interest.

## Nomenclature

| | |
|---|---|
| GHG | Greenhouse gases |
| CI | Carbon Intensity |
| EU | European Union |
| Solar PV | Solar Photovoltaic |
| RES | Renewable Energy Resources |
| NREL | National Renewable Energy Laboratory |
| HOMER | Hybrid Optimization Model for Multiple Energy Resources |
| LCOE | Levelized Cost of Energy |
| EV / EVs | Electric Vehicle/Electric Vehicles |
| DHW | Domestic Hot Water |
| NPC | Net Present Cost |
| IRR | Internal Rate of Return |
| GEWEX | Global Energy and Water Exchange |
| SRB | Surface Radiation Budget |
| O and M | Operation and Maintenance Cost |
| GMAO | Global Modelling and Assimilation Office |
| MERRA-2 | Modern Era Retrospective-Analysis for Research and Applications |
| FP-IT | Forward Processing—Instrument Teams |
| IGCC | Integrated gasification combined cycle |
| LHV | Lower Heating Value |

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
