# Peer review of "Assessment of a Fully Renewable System for the Total Decarbonization of the Economy with Full Demand Coverage on Islands Connected to a Central Grid: The Balearic Case in 2040"

_machines, doi:10.3390/machines11080782_

Round 1

Reviewer 1 Report

Dear authors,

I am glad to have the opportunity to read such an interesting paper, even though I have a few suggestions to improve the quality of the paper, these are:

1-Consider including this paper in your manuscript as it matches your study https://doi.org/10.1016/j.scs.2022.104375

2-Line 73, remove extra points

3-I wonder beyond the results that HOMER provides you, which is your contribution to the state of the art?

4-I also wonder, how the match between peak demands and peak generation is analysed in this study, since is well known that these islands are overcrowded during the Summer season. 

5-In section 2, I found that you state in line 129 10.3 TWh per year, but in Fig 1, you declare 10.2 and also, when I make the sum and as a result, I obtained 10.4, so check this out.

6-Reference this argumentation "The EV already emits in average 55% less than conventional vehicle for sustainable economies"

7-Reference this argumentation, "In hotels and accommodations, air conditioning and DHW account for 50-70% of consumption." What about heating systems, why is not bear in mind?

8-Section 2 lacks brightness although it contains a lot of information is an endless list of bullets that for an article it looks null, it might be improved.

9-Why use HOMER and not other software? please consider this article to be part of your manuscript "https://doi.org/10.1016/j.rser.2019.109691"

10- In Figure 3 you assume that the load of that island is totally RESIDENTIAL and it is not true, this load is typically presented for domestic demands. Additionally, you are assuming that the EV mainly recharge in the super valley period, why for cost reasons or for time matters?

11-Please, reference the O&M cost I miss it in the paper. 

12-Table 5 has vanished (I mean line 612). The figures are a bit messy, you cite figures 10,11, and 12, while you are presenting figure 7, it is difficult to follow. I kindly recommend you explain each figure more in detail immediately after is shown.

A deep revision of the grammar is highly advisable and needed.

Author Response

Dear reviewer, 

Our reply is presented in the attached document.

Thanks in advance,

Dr. Paula Bastida Molina (corresponding author)

Reviewer 2 Report

Firstly my main remark is related to the size of this paper. In my opinion it consists of to many pages, what does not means that it is not interesting. I’m wonder how the Authors obtain the data for their analyzes for 2040. It is also not clearly presented in the paper. I’m also not sure what exactly looks the system of the energy production and what scenario was used by the Authors. I mean what other system of energy production and scenarios ware taken into account (if) and why Authors selected this one. As one of the novelty of this paper Authors emphasized its electric grid interconnection with mainland while others considered only off grid system. However in the paper I didn’t find information how this interconnection will be use. Only for the selling the surplus of produced energy? why it can’t be used also for the energy deficiency balance? Are the EV batteries may be used as the part of storage system, thanks that the size of battery storage system may be smaller?

In the case of energy produced by the PV panels it should be also estimated the influence of the temperature on the PV efficiency, because the annual temperature rise according to the climate change

Edistors remarks

Rows 612 613 needs to be edited 

629 there is some text highlighted in yellow 

640 4,5*105 the last 5 is nomial as I understand 

629 to 644 is the same as 680 to 695

In row 645 Table 7 is wrong table because it is related to the hydrogen production.

Figure 7, however interesting, for me is completely not clear. It is impossible to see the distribution during the one year of the values. This represents the data or simulation results or prediction? The same Figure 8

Author Response

(The authors gave the same response as above.)

Reviewer 3 Report

  1. The content on page 15, line 612, of Table 5 is missing.
  2. The reference resources for "For this reason, it has been considered that hydrogen demand can vary by up to 10% between consecutive days" should be clarified. And under the very worst weather conditions, the variation could be above 10%, and the related effects on the system should be discussed.
  3. Those results generated based on HOMER optimization should be clarified. And the related simulation steps based on HOMER should be clarified.
  4. The PV capital cost in Table 8 should be detailed based on the data resources and the calculation details based on its capacity. It is much higher compared with the wind capital cost. And the related MW of each system should be listed in Table 8.
  5. The contributions of this work should be listed clearly in the Introduction section.

Author Response

(The authors gave the same response as above.)

Reviewer 4 Report

Your investigation work is fascinating; the simulation and discussion are solid and consistent. There are only a few details that could be improved:

1. Do a general language and spelling check. There are some errors throughout the text.  For a quicker correction here are the lines and mistakes you may check.

 line 154 figure and units of the amounts should appear together with the years

line 161 reviewing the nomenclature

line 294 check spelling and punctuation

Adding a section with the abbreviations is recommended as many are also used and it is not easy to read.

table 2, table 3, and table 4  could be rearranged to the center.

correct spaces in lines 612-613, 710, 734

unify bold on lines 623 and 629, 645

correct nomenclature (subscript) in lines 632, 635, 638, 640…

checking figure title online 772

unify spaces in quantities either with or without (for example 787 or 966, 972)

Author Response

(The authors gave the same response as above.)

Reviewer 5 Report

The paper content is very interesting. However, there are some points where it must be improved. Please consider the following suggestions.

1.       Line 72, “air transport...” the dots are incorrect

2.       I suggest to include the source of Figure 1 in the legend

3.       Figure 1 content is not clearly explained; I suggest to redesign it or explain it more extensively.

4.       Lines 141-283. It is not clear where this information come from; I suggest to explain more extensively the method you used to extract these.

5.       Both figures 1 & 2 resolution is not clear. I suggest to improve it.

6.       Line 612 – has to be corrected

7.       There are abbreviations i.e. O&M that are not explained. I suggest to explain them where they are first met.

8.       It is not clear to the reader how the scenario was analysed was chosen; I suggest to explain the rationality of the selection more extensively.

9.       Paragraph 5 is named Results and discussion as 5.5, this is confusing; I suggest to rename them in order to be clear what is the content of each one.

There are several linguistic mistakes and language improvement is needed. I suggest the authors to review English language and style once again.

Author Response

(The authors gave the same response as above.)

Round 2

Reviewer 1 Report

Dear Authors,

Once the manuscript has addressed all my kind suggestions and comments I find it has enhanced the readability and the quality. So, I strongly support it for publishing as it is currently.

Congratulation on such an interesting analysis.

Sincerely

Author Response

Dear reviewer,

Thank you very much for your review.

Best regards,

Paula Bastida-Molina (corresponding author)

Reviewer 5 Report

Dear authors,

The paper has been significantly improved. However, there are some points that could be further cleared. Please consider the following suggestions.

1.       Line 143- 145, “For these uses, hydrogen currently seems to be the most advantageous alternative, which means approximately 50·103 tons of H2 per year to cover the forecasted 1.7 TWh of non-electrifiable energy end-uses.” I suggest to state the source of this information and how this result occured.

2.       The title “4.3. Discussion and main conclusions of the Balearic Islands electricity system” is still confusing. I suggest to consider name this paragraph ‘findings’ or something similar and include a separate standalone discussion paragraph.

Author Response

Dear reviewer, 

Our reply is presented in the attached document.

Thanks in advance,

Dr. Paula Bastida-Molina (corresponding author)
